# Relationships between the Content of Micro- and Macroelements in Animal Samples and Diseases of Different Etiologies

**DOI:** 10.3390/ani13050852

**Published:** 2023-02-26

**Authors:** Marina V. Stepanova, Larisa F. Sotnikova, Sergei Yu. Zaitsev

**Affiliations:** 1Department of Bioecology and Biological Safety, Federal State Budgetary Educational Institution of Higher Education “Russian Biotechnological University (ROSBIOTECH)”, Volokolamsk Highway, 11, 125080 Moscow, Russia; 2Research Laboratory of Ophthalmology, Oncology and Biochemistry of Animals, ROSBIOTECH, 125080 Moscow, Russia; 3Department of Diseases of Small Domestic and Exotic Animals, ROSBIOTECH, 125080 Moscow, Russia

**Keywords:** animal health and diseases, essential and toxic microelements, classes of diseases, general morbidity of animals, atomic absorption spectrometry

## Abstract

**Simple Summary:**

Macro- and microelements (MMEs), simply called as “minerals”, are essential biogenic elements for animals and humans. These two MME groups are very important and differ only in that the first group) macroelements (Ca, P, K, Na, S, Cl, Mg) are containing in concentrations more than 0.01% and required by the body in doses from a few grams to several milligrams per day, whereas the second group) microelements (Fe, Zn, Cu, Mo, Mn, I, etc.) – from ten to hundred times less. MMEs are distributing in all the tissues and organs, playing the key roles in their functions. The numerous enzymes (as protein biocatalysts) are using MMEs as essential elements (“cofactors” or “coenzymes”) for metabolic reactions in all cells. Some researchers, including us, are mentioning toxic heavy metals (Arsenic, Cadmium, Lead, Mercury, etc.) in the MME general concept, but as the third special group, because these metals are quite dangerous and have the ability to cause various diseases (cardiovascular, metabolic, nervous, oncological diseases, etc.). It is important to monitor the MME-status of the animals regularly by using non-invasive biological materials (such as hair, fur, etc.).

**Abstract:**

Many of the micro- and macro-elements (MMEs) required by the body are found in environmental objects in concentrations different from their original concentration that can lead to dangerous animal diseases (“microelementoses”). The aim was to study the features of MME (accumulating in wild and exotic animals) in connection with particular diseases. The work using 67 mammal species from four Russian zoological institutions was completed in 2022. Studies of 820 cleaned and defatted samples (hair, fur, etc.) after “wet-acid-ashing” on an electric stove and in a muffle furnace were performed using a Kvant-2A atomic absorption spectrometer. The content of zinc, copper, iron, cadmium, lead, and arsenic was assessed. The level of MME accumulation in the animal body contributes not only to the MME status and the development of various concomitant diseases, but the condition itself can occur by intake of a number of micronutrients and/or drugs. Particular correlations between the accumulation of Zn and skin, oncological diseases, Cu—musculoskeletal, cardiovascular diseases, Fe—oncological diseases, Pb—metabolic, nervous, oncological diseases, and Cd—cardiovascular diseases were established. Therefore, monitoring of the MME status of the organism must be carried out regularly (optimally once every 6 months).

## 1. Introduction

Many “micro- and macro-elements” (MMEs) are found at different concentrations in environmental objects (especially in areas with a high anthropogenic load) that can lead to animal diseases [1,2,3,4,5,6]. The supply of microelements to all organisms should occur only in optimal amounts [1,7,8,9,10]. This is well-known for all domestic [11,12,13,14] and farm [15,16,17,18,19,20,21,22,23,24] animals, as well as for humans [25,26,27,28]. That is why here we would like to point out only a few recent reviews concerning the comparison of inorganic and chelate forms of MMEs for broilers’ feeding [29] and some requirements of MME diets for two commercial broiler strains [30]. For example, the numerous data of the most important MMEs (such as iron, copper, zinc, manganese, selenium) indicated that the chelate forms of these MMEs (i.e., complexes of MME with organic ligands) provided better “protection” for broilers, for “the environment and also improve egg quality” [29].

A substantial comparison of the 9th and 10th versions of nutrient requirements of poultry (NRC 1994 and 2017) with the particular recommendations of “the trace element requirements for commercial broiler strains” (Ross 308 and Cobb 500, since 2007 and 2008, respectively) was provided by Iran scientists [30]. In the “Conclusion” part of this paper [30], they wrote “the iron requirements of broilers have been increased and the requirements of copper, manganese, zinc, selenium and iodine have decreased compared with the NRC (1994) recommendations”. However, all data (in the paper [30]) presented the opposite tendencies. The iron requirements have been decreased from 80 mg/kg (NRC 1994 for broiler chickens) to 40 mg/kg (for broiler strains Ross 308, since 2007 and Cobb 500, since 2008 or 2013) or even to 20 mg/kg (for broiler strain Ross 308, since 2014). The requirements of copper, manganese, zinc, selenium, and iodine have been increased from 8, 60, 40, 0.15, and 0.35 mg/kg (NRC 1994 for broiler chickens) to 16, 120, 100–110, 0.30, and 1.25 mg/kg (for broiler strain Ross 308, since 2007–2014) or 15, 100, 100, 0.30–0.35, and 1.0 mg/kg (for broiler strain Cobb 500, since 2008–2013) [30].

In general, less is known about MMEs in the tissues of the wild and exotic animals compared to those for the farm animals and birds. It is especially important now because of the high technological activity all over the world that results in the introduction of the increasing amounts of various “trace elements into biogeochemical cycles” [25].

In recent decades, wild animals have been successfully and increasingly used as “bioindicators of environmental pollution” [31,32]. Due to their wide distribution and often high trophic levels, birds and mammals are the most suitable indicators of pollution, and therefore the determination of trace element concentrations, including heavy metals, in various tissues of different bird and mammal species (for example, in blood, feathers, liver, kidneys, muscles, eggs, feces, etc.) is widely used in “biomonitoring studies” [33]. However, the most prominent studies [19,20,21,22,23,24] have been connected with the exposure of animals to pollution in heavily polluted areas or aquatic ecosystems, whereas the relatively moderately polluted urban and suburban environments have been less studied, despite their ecological significance [34]. Most studies are aimed at elucidating the effect of one pollutant or elevated doses of several substances [35,36], but there are still no clear patterns concerning the effects of trace elements in animals.

A number of Polish authors [24,34,37] revealed significant differences in the level of accumulation of microelements in animals living in nature and kept in captivity (i.e., in urban facilities). Therefore, the study of the features of their accumulation in animal zoological institutions, taking into account the characteristics of their maintenance and nutrition, is significant [37].

It is important to highlight that a deviation from the optimal level can lead to the animal diseases (“microelementoses”) with various degrees of severity [1,27,28].

### 1.1. MMEs in Animal Tissues and Organs

Animals need MMEs in certain concentrations for the implementation of optimal metabolism level in their tissues and organs, as well as in the whole body. The clinical picture of the pathology occurrence (taking into account the complex interaction between individual microelements, the physical and chemical properties of their compounds) leads to difficulties of data interpretation due to the presence of direct and indirect (not always explicit) effects [1,9,10,13]. For these purposes, it is of great importance to choose an adequate (“correct”) marker, i.e., “biosubstrate”, and comprehensive monitoring of the MMEs level, which will allow their accurate quantitative analysis [1,31,38,39].

Each MME has its own optimal level of content depending on the animal breed. If this level deviates towards an increase or decrease in the concentration of an element in the body, a biological effect occurs. Moreover, the degree of bioeffect development depends on the microelement dose. Acute poisoning develops when the body absorbs or accumulates relatively large doses of toxicant MME. Chronic poisoning develops with prolonged, cumulative, action of chemicals. In this case, the increase in symptoms and the clarity of the manifestation of the clinical picture occurs gradually. The contact of chemical compounds with the epithelium and skin leads to their entry into the blood and lymph. Absorption is accompanied by special “transformation” of compounds, distribution and accumulation of microelements by organs and tissues [40].

Four major and minor zoos in Russia were selected for our research.

### 1.2. The Moscow Zoo

The Moscow zoo is the first among the major zoos in Russia and one of the oldest zoos in Europe, which was opened on February 13 (January 31), 1864. Now, it has the largest zoological collection in Russia and is the leading methodological center in the system of zoos in our country. At the moment, this zoo includes: a center for the reproduction of rare species of animals (zoo nursery) of the Moscow State Zoological Park, a branch of the zoo in the Estate of Father Frost (“Veliky Ustyug”), and the main exposition parts in the center of Moscow city. In total, this zoo contains 1340 species of various breeds and exhibits 14,105 wild and exotic animals, birds, etc., in total [41]. The research facility of this institution specializes in breeding rare species of predatory mammals, birds of prey and waterfowl, cranes, poisonous snakes, etc. [41,42].

There are about 15% of mammals in the total number of animals in the institution collection and 24–26% of birds (depending on the studied period) [41,42]. The change in the number of animals is associated mainly with the acquisition and exchange of species between zoological institutions and the reproduction of formed pairs. During the studied period, the total collection of mammals increased by 19% of species (from 174 to 207) and 8.7% in total head numbers (from 1462 to 1589 items) mammals; birds increased by 12.3% of species (from 292 to 328), but the total head numbers decreased by 0.2% (from 620 to 612 items) [41,42,43].

### 1.3. The Ivanovo Zoo

Ivanovo zoo is a rather small institution located in the capitol of the textile regional center (officially opened in 1994). Today, the territory of the zoo is about 3.4 hectares. In total, this zoo contains 790 animals of 173 species. The zoological institution specializes in breeding birds of prey. The collection of the institution is represented mainly by birds (60.8–67.6%) and mammals (22.0–28.9%) [41,42]. In this zoo, the species diversity of mammals decreased by 25.5% (from 51 to 38 species) and by 13.7% in numbers (from 256 to 221 items). The number of bird species and heads increased by 9.3% and 12% (from 107 to 117 species and from 475 to 532 items, respectively) [41,42,43].

### 1.4. The Yaroslavl Zoo

Yaroslavl zoo is the first landscape-type zoo in Russia, where animals are kept in conditions as close to natural as possible. Today, the territory of the zoo is about 123 hectares (under the exposition—58 hectares). In total, 1791 animals of 441 species live in this zoo. The institution specializes in keeping and breeding animals of the central Russia population. The collection of the institution is represented by birds (18.3–20.8%) and by mammals (13.0–14.9%), depending on the year of the study [41,42]. The species diversity of the mammalian exposition decreased by 11.5% species (from 104 to 92), but slightly increased by 0.5% in total head numbers (from 440 to 442). The number of bird species and heads decreased by 1.3% (from 160 to 158 and from 620 to 612 items, respectively). The change in the number of livestock is associated mainly with the physiological conditions of some old animals [41,42,43].

### 1.5. The Uglich Zoo Station

The Uglich zoo station is one of the smallest in Russia and located in a historical building since 1936. During the last year, more than 5000 people from the region and city guests came to the station for excursions and public events. In total, this station contains 305 individuals belonging to 52 animal species. The institution specializes in keeping and exhibiting exotic and domestic animals (especially in the contact mode). The collection of the institution, depending on the year of the study, is represented by birds (18.2–28.8%) and by mammals (25.0–31.9%) [41,42]. The species diversity of the mammal exposition remained at the level of 13 species, but total head numbers increased by 33.7% (from 89 to 119 items, due to exchange and birth) [41,42]. The number of bird species and heads increased by 87.5% and 77.3%, respectively (from 8 to 15 species and from 22 to 39 heads) [41,42]. The change in the number of livestock is associated with the organization of regular monitoring and veterinary care of animals [41,42,43].

The aim of this study was to evaluate the features of the MME accumulation in the biological samples (hair, fur) of wild and exotic animals from these four zoos during some diseases of various etiologies.

## 2. Materials and Methods

### 2.1. Living Creatures as Objects

The studies were carried out based on the Moscow, Ivanovo, Yaroslavl, and Uglich zoos. The objects were wild and domestic animals of different taxonomic groups.

The studies were carried out during the whole year. The level of microelements, including heavy metals, in biological media was analyzed based on the results of our original research. The following species of wild, domestic, and exotic mammals kept in zoological institutions in territories with different anthropogenic pressure in the Central Federal District of Russia were selected for study: Moscow, Ivanovo, and Yaroslavl zoos. In particular: bristly armadillo—*Chaetophractus (Euphractus) villosus* (n = 15), globular armadillo—*Tolypeutes matacus* (n = 9), Egyptian flying dog—*Rousettus aegyptiacus* (n = 15), hare—*Lepus timidus* (n = 9), European hare—*Lepus europaeus* (n = 6), yellow pied—*Eolagurus luteus* (n = 15), Mongolian (clawed) gerbil—*Meriones unguiculatus* (n = 36), Bush-tailed gerbil—*Sekeetamys calurus* (n = 18), Eastern mole vole—*Ellobius tancrei* (n = 6), Cactus hamster—*Peromyscus eremicus* (n= 9), golden (Syrian) hamster—*Mesocricetus auratus* (n = 30), Djungarian hamster—*Phodopus sungorus* (n = 15), Baraba hamster (Chinese hamster)—*Cricetulus barabensis griseus* (n = 9), acacia rat—*Thallomys loringi* (n = 15), spiny mouse—*Acomys cahirinus* (n = 30), dwarf mouse—*Mus minutoides* (n = 9), gray rat—*Rattus norvegicus* (n = 18), house mouse—*Mus musculus* (n = 18), multi-mother mouse—*Mastomys natalensis* (n =9), chinchilla (home form)—*Chinchilla laniger var. dom.* (n = 18), degu—*Octodon degus* (n = 24), guinea pig—*Cavia porcellus* (n = 27), Indian porcupine—*Hystrix indica (leucura)* (n = 12), red fox—*Vulpes vulpes* (n = 18), Fennec fox—*Vulpes (Fennecus) zerda* (n = 6), Arctic fox—*Alopex lagopus var. dom.* (n = 12), Alaskan Malamute—*Canis familiaris* (n = 12), polar wolf—*Canis lupus tundrorum* (n = 6), wolf—*Canis lupus* (n = 39), raccoon dog—*Nyctereutes procyonoides* (n = 15), raccoon—*Procyon lotor* (n = 6), nosoha—*Nasua nasua* (n = 9), domestic ferret (furo, ferret)—*Mustela putorius var. Dom.* (n = 15), common genet—*Genetta genetta* (n = 9), brown bear—*Ursus arctos* (n = 12), Ussuri white-breasted (Himalayan) bear—*Selenarctos (Ursus) thibetanus ussuricus* (n = 6), lynx—*Felis (Lynx) lynx* (n = 15), puma—*Puma (Felis) concolor* (n = 9), snow leopard (Irbis)—*Uncia (Panthera) uncial* (n = 6), Far Eastern (Amur) leopard—*Panthera pardus orientalis* (n = 9), Amur tiger—*Panthera tigris altaica* (n = 12), white lion—*Pantera leo var. alba* (n = 6), lion—*Panthera leo* (n = 6), Bruce’s hyrax—*Heterohyrax brucei* (n = 3), alpaca—*Vicugna pacos* (n = 12), bactrian camel—*Camelus bactrianus (ferus) dom.* (n = 21), camel—*Camelus dromedarius* (n = 6), reindeer—*Rangifer tarandus* (n = 28), spotted deer—*Cervus nippon* (n = 9), European fallow deer—*Dama (Cervus) dama* (n= 9), European elk—*Alces alces* (n = 18), musk ox—*Ovibos moschatus* (n = 6), domestic yak—*Bos mutus (grunniens) var. dom.* (n = 12), bison—*Bison bonasus* (n = 9), Dagestan tur—*Capra cylindricornis* (n = 9), Sichuan takin—*Budorcas taxicolor tibetana* (n = 6), blue sheep—*Pseudois nayaur* (n = 3), Cameroonian goat—*Capra hircus hircus* (n = 12), domestic horse—*Equus caballus* (n = 33), Grant’s zebra—*Equus burchelli boehmi* (n = 6), Przewalski’s horse—*Equus przewalskii* (n = 9), domestic donkey—*Equus asinus dom.* (n = 12), common marmoset (Wistity)—*Callithrix jacchus* (n = 6), ring-tailed lemur—*Lemur catta* (n = 6), mandrill—*Mandrillus (Papio) sphinx* (n = 6), rhesus monkey—*Macaca mulatta* (n = 6), and lapunder (pig-tailed macaque)—*Macaca nemestrina* (n = 9).

### 2.2. Methods

The studies were carried out on a “Kvant-2A” atomic absorption spectrometer. The selection of biological samples (hair, fur, etc.) of all types was carried out from the whole body with a total sample weight of about 1–3 g. The samples were cleaned and degreased with acetone and bidistilled water for two days. Then, “wet-acid-ashing” was carried out on an electric stove, and then in a muffle furnace with a gradual increase in temperature from 250 to 450 °C with a half-hour exposure. The samples were assessed for the level of trace elements—zinc, copper, iron, cadmium, lead, and arsenic.

The results obtained were processed statistically. Arithmetic mean values (M), mean errors (m), and standard deviation (δ) were determined. To identify statistically significant differences in the compared groups and the contingency between the signs, the nature of the distribution of compatibility data, the nonparametric criterion (W criteria, Shapiro–Wilk test), Student’s *t*-test, and Spearman’s correlation coefficient were used. Databases were formed in the programs “Microsoft Office Excel” 2010 and “Statistica” version 10.0 (Windows XP).

## 3. Results and Discussion

As a first part of this work, a study of the nosological profile of diseases of wild and exotic birds and mammals of zoos (the Yaroslavl, Moscow, Ivanovo, and Uglich zoos) was carried out.

### 3.1. The Moscow Zoo

The annual mortality in the institution was about 0.3% of all animal specimens: birds—0.4–0.5%; mammals—0.2% [41,42]. In the studied period, parasitic diseases were recorded. Most enclosures with predatory mammals are permanently unfavorable for ascariasis and toxocariasis. It is impossible to carry out a complete devastation of open enclosures, because a part of the development cycle of ascariasis and toxocariasis occurs in the soil, where eggs can be stored for months. For young and healthy animals, carriage is not dangerous and, as a rule, does not affect their condition. However, in aging or chronically ill animals, the presence of parasites can cause various clinical symptoms. A similar situation develops with helminths in ungulates—”strongyloidosis” and “ascariasis” were recorded. Animals are constantly dewormed and the degree of infection is maintained at a “safe level”. Various types of nematodes were observed in almost all groups of birds living in outdoor enclosures, which is associated with the presence of synanthropic birds in enclosures. Some populations of parrots turned out to be a reservoir of megabacteriosis. There was a trend towards an increase in the number of gerontological diseases (chronic diseases of the musculoskeletal system, cardiovascular system, chronic kidney diseases, and tumors for aged animals), which is associated with the aging of the collection. It should be noted that one of the systemic problems is obesity, especially for mammals (diagnostic studies) [41,42,43].

### 3.2. The Ivanovo Zoo

During the year, up to 9.5% of the animals from the total number of livestock were exposed to diseases. The survival rate of animals (after past diseases) was 94.7% in all cases. Of the total number of established diseases of animals, 50.7% were injuries, 17.3%—diseases of the respiratory system, and 16.0%—diseases of the digestive system. Then, other diseases were observed in 8.0% cases, metabolic disorders in 6.7% cases, and diseases of the reproductive organs in 13% cases [41,42,43].

### 3.3. The Yaroslavl Zoo

About 12.8% of animals from the total number of livestock were exposed to diseases during the year [41,42]. There is a regular increase in registered diseases by 34.2–76.3%, which was associated with an increase in the number of livestock and an improvement in the results of diagnostic tests due to the purchase of laboratory equipment. The effectiveness of the therapeutic measures taken was confirmed by the increase in the survival rate of animals after diseases from 55.4% of cases to 79.3% in recent years. Of the total number of established diseases of animals—31.2% were diseases of the digestive system, then we observed lesions of the musculoskeletal system—17.3% and the cardiovascular system—10.1%, diseases of the hearing organs and the nervous system are sporadically noted—1.1% and 1.3%, respectively [41,42]. Cancer diseases were detected in 5.05% of individuals during the year [41,42]. The main causes of diseases of non-contagious etiology in wild animals in zoos were the factors that limit their active movement and constant stress factors due to the specifics of the institution [41,42,43].

### 3.4. The Uglich Zoo Station

During the year, up to 37.4% of animals from the total number of livestock were exposed to diseases. The percentage of survival of animals (after past diseases) was about 41.9% in all cases [41,42]. Of the total number of established animal diseases, 45.3% occurred against the background of physiological conditions of some old animals (since most often individuals came to the institution from visitors), 24.4% were diseases of the digestive system, 11.08% were lesions of the musculoskeletal system, in 7.2% had diseases of the respiratory system [41,42,43].

### 3.5. A General Description of Animal Diseases

It was found that up to 12.8% of animals from the total number of livestock were subjected to various diseases. Of the total number of established animal diseases, the main share was diseases of the digestive system, then lesions of the musculoskeletal system and the cardiovascular system. Moreover, diseases of the hearing organs and the nervous system were sporadically noted [41,42]. During the research, an increase in the proportion of oncological diseases in all the studied institutions was established [41,42,43].

Based on a retrospective analysis of records entered in the register of sick animals, 1208 heads of wild animals were treated, i.e., about 12.9% of animals from the total number of livestock. Clinical examination was carried out as follows: history taking, general clinical examination, laboratory tests. The information obtained was recorded in the medical history of each animal.

Skin diseases (dermatitis, urticaria, fungal infections) accounted for 2.3–2.5% of the total number of non-communicable diseases. A significant increase in the number of animals with pathology of the hearing organs (otitis media, injuries) was observed, amounting to 2.5%, diseases of the musculoskeletal system in wild animals—21.6%. Most often during the study period, various traumatic injuries were noted that occurred in animals and birds as a result of intraspecific and interspecific aggression, a lot of injuries were observed in mixed species exposures. Some injuries that the animals inflicted on themselves on the structural elements of the enclosures were revealed.

Respiratory system diseases were observed in 6.2% of cases, among them inflammatory pathologies were noted—acute and chronic pneumonia and bronchopneumonia (of unknown etiology), rhinitis, tracheitis.

At the same time, diseases of the digestive system of non-contagious etiology were registered in 25.6% of the total number of non-contagious animal diseases. The main disorders of the gastrointestinal tract, as a rule, were interconnected with a violation of the functional state of the liver; hepatosis (fatty, granular, cholestatic) was established in 33.1% of cases of diseases, then enteritis (17.5% of diseases) and poisoning, 6.8%; intestinal obstruction was established in isolated cases only.

The level of metabolic diseases during the research period was found between 4.8 and 4.9% of cases, including exhaustion and impaired calcium-phosphorus metabolism. During the research period, diseases of the cardiovascular system accounted for 10.2%, detected mainly at the autopsy of animals, and a high percentage of pathologies of the heart muscle was noted—25.8%. Diseases of this system were represented by atherosclerotic changes in the coronary arteries, myocardial infarction, hypertrophic cardiomyopathy, and myocardial dystrophy. Inflammatory diseases were represented by chronic lymphocytic pericarditis and chronic focal myocarditis (complicated by thrombosis of the right ventricle—in the fluffy-tailed gerbil), cardiomegaly myocardial dystrophy, as well as a case of hemopericardium of unknown etiology. The reasons may be inadequate feeding with a deficiency in the diet of carbohydrates and minerals, autointoxication from the liver and kidneys.

Eye diseases were established in 8.0% of cases. Basically, conjunctivitis was observed, which is associated with a violation of the conditions of detention, for example, abundant contamination of the litter, strong dust formation due to too small a fraction of sawdust. The decrease in the level of morbidity was associated with the formation of new expositions, the operating time of suppliers, and the improvement of the skill level of service personnel.

Among the diseases of the musculoskeletal system, the most common were arthritis, bursitis, discopathy, myositis, sprain, inflammation of the jaw bone plate, and various kinds of injuries.

The incidence of the reproductive system was detected in 6.3% (dystocia, ovarian cyst complicated by secondary infection and abscess formation, follicular stasis). Often there was a pathology of labor and postnatal activity in ungulates, a delay in the placenta was observed. Predisposing factors for this may be multiple pregnancies (twins), long, difficult births, and lack of vitamins and minerals in the concentrated type of feeding. The increase in the number of diseases was associated with the formation of breeding pairs in the institution and the entry of young animals into the sexually mature stage.

Insignificant fluctuations in the incidence of the nervous system during the study period from 1.4 to 1.6% were noted. The low level of diseases of the nervous system was explained by the presence of large open-air cages, the formation of species groups, mixed species exposures, and regular enrichment and diversity of the animal habitat.

During the research period, diseases of the excretory system were detected in 9.5% of the total number of diseases (renal failure was due to various causes: chronic glomerulonephritis with outcome in nephrosclerosis, hydronephrosis, pyonephrosis, acute interstitial nephritis; unspecified nephropathy, ascites). The cause of kidney pathology may be an unbalanced diet, the presence of mycotoxins in feed.

Oncological diseases accounted for 13.8% of the total number of diseases and 19.5% of the number of non-communicable diseases. The most common malignancies were carcinomas and adenocarcinomas.

Based on the results obtained, a model of the nosological profile of non-communicable diseases (of exotic, wild, and ornamental animals kept in captivity) was obtained. Based on the nosological profile of non-communicable diseases, one can conclude that the most often there are the following diseases: of the digestive system—in 31.2% of cases, of the musculoskeletal system—17.3%, of the cardiovascular system—10.1%, of the nervous system—1.3%, and of the hearing organs—1.1% for exotic, wild, and decorative animals.

To establish the relationship between the MME content in samples of biological media of animals with general morbidity during the study period, a correlation-regression analysis was carried out, the results of which are presented in Table 1. The diagnosis was established based on the study of individual animal maps, pathological anatomical acts, and registers of veterinary procedures that was proposed before [44,45,46,47,48,49,50,51]. 

A significant relationship was established between the accumulation of Zn and the following diseases: skin, digestive, and vision systems, as well as oncological diseases, Cu—with diseases of the musculoskeletal and the cardiovascular system, oncological diseases, Fe—with diseases of the cardiovascular system, Pb—with metabolic diseases, nervous and excretory systems, oncological diseases, Cd—with diseases of the cardiovascular and nervous system, and As—with diseases of the excretory systems (Table 2), which corresponded to some other data [1,42,52,53].

### 3.6. Zinc

Pairwise statistical analysis of the data (Table 2) revealed a significant (*p* < 0.05) decrease in the level of Zn in animals with diseases of the organs of vision, digestive system, and skin diseases. There was also a tendency of a decrease in the zinc concentration in the case of metabolic disorders and diseases of the reproductive system [53].

Zinc is believed to be essential for optimal retinal cell metabolism, modification of photoreceptor plasma membranes, regulation of the light-rhodopsin response, and modulation of synaptic transmission. In mice, the ZnT3 and ZnT7 transporters present in different layers of the retina, along with MME, provide zinc homeostasis and are involved in the functions of the eye [54]. Moreover, zinc is required to maintain the level of taurine in the retina by acting on the transporter responsible for the movement of taurine into tissues [55]. In animals, the effect of low zinc levels on abnormalities in the functioning of photoreceptors, the onset of primary glaucoma, disorganization of the ultrastructure, and loss of color sensitivity has been shown [56,57].

A decrease In the level of Zn in the digestive system may be associated with a decrease in food intake due to a decrease in animal weight or the development of various infections, diarrhea, which prevents MME’s reabsorption [58,59].

Skin diseases with a decrease in zinc levels may be associated with a violation of its homeostasis with a decrease in the zinc transporter present in epidermal skin cells, etc. (Figure 1) [60,61,62,63].

For example, ZIP4 and especially ZIP2 are highly expressed in keratinocytes and are involved in their proliferation [61]. Moreover, ZIP10 is highly expressed in epidermal progenitor cells located in the outer root sheath of hair follicles and plays an important role in the regulation of zinc to maintain the skin epidermis [62]. In vitro studies have shown that ZIP2 is activated by the induction of differentiation in cultured keratinocytes and that when this transporter is disabled, differentiation is inhibited [63].

The effect of Zn on the excretory system may be due to the fact that the metal mainly accumulates in the liver [8,13,64,65], while the biliary system is the main pathway zinc excretion and enterohepatic circulation [66]. A number of authors note a decrease in the level of the microelement in pathological lesions [13] associated with the presence of metallothionein [66], which is regulated by a decrease in the level of intake of food and water in the body with low body weight caused by liver disease, cytokines and inflammatory mediators, general inflammation, stress, and medications such as glucocorticoids [66].

The importance of zinc for reproductive function in males has been documented [67]. Zinc ions in the seminal fluid play both structural and regulatory roles in the activity of prostate-specific arginine esterase, which maintains normal prostate and sperm function [68]. For females, data on the effect of Zn on reproductive function are practically absent, there are only indications of its significance in the formation and development of pregnancy and fetal size [69,70]. There are observations of a significant increase in the concentration of Zn in both malignant and benign tumor skin tissues [71]. This was due to two possible reasons: intensive metabolic processes in neoplastic cells and increased activity of intracellular enzymes that require intracellular zinc for proper functioning, or an increase in intracellular Zn, which inhibits tumor cell apoptosis [71]. In contrast, zinc was significantly lower and copper significantly higher in neoplastic tissue with hepatocellular carcinoma [72] and females with mammary tumors [73] compared to healthy tissues. This can be explained by zinc chelation, which is especially important in this context, since copper is essential for angiogenesis, and by reducing tissue copper levels, zinc can limit tumor growth [74]. In the study, there were no significant differences in the accumulation of MME in the case of some oncological diseases.

### 3.7. Copper

During the statistical analysis of the data, a significant (*p* < 0.05) decrease in the Cu level for animals with a disease of the musculoskeletal system was found. In contrast, an increase in the Cu level by oncological disorders and diseases of the cardiovascular system was established.

Although a deficiency of copper, which is involved in normal iron transport, may be accompanied by the development of iron deficiency anemia [75], its excess can lead to antagonism between the both metals due to the presence of common transport pathways, primarily the DMT-1 protein (divalent metal transporter 1, Figure 2) [76].

In particular, an increase in the level of copper in the hair of patients with iron deficiency anemia has been noted [77].

It has been established that copper plays an important role in cancer cell proliferation and metastasis [78], since cancer cells are characterized by a high demand for copper [79]. The impact on the mobility of copper ions can be one of the tools of anticancer therapy [79]. At the same time, an increased concentration of copper in the blood was found in patients with pancreatic cancer [80]. There are indications that high levels of copper in hair are associated with the risk of developing prostate cancer [81], as well as neoplasms in children [82]. At the same time, it is noted that under the conditions of the physiological level of copper intake into the body (0.6–3 mg/day), the existence of a relationship between copper in the body and oncology is unlikely [83].

These results, in relation to Cu, can be explained by the fact that the deficiency of this MME leads to deficient collagen synthesis, accompanied by skeletal deformity, and changes in elastic fibers, which further leads to the occurrence of orthopedic diseases or Cu deficiency in the body is a consequence of such a disease [1].

### 3.8. Iron

A significant decrease in the level of iron was found by development of cardiovascular diseases. Dysregulation of Fe homeostasis, increased uptake, and accumulation of MME in the reticuloendothelial system leads to the removal of the element from the blood into the cells of the reticuloendothelial system following a decrease in the availability of iron for erythroid progenitor cells and iron-limited erythropoiesis [84]. The acute phase protein hepcidin plays a key role in the development of anemia due to its ability to inhibit Fe absorption in the intestine. In addition, at the same time, there is an increase in the uptake of iron by macrophages and blocking of the export of iron from macrophages, mainly to the bone marrow. As a result, serum iron concentration decreases (with normal total body iron), which slows down erythropoiesis and causes anemia [85]. However, sometimes such a drop in serum iron concentration can be beneficial, as it makes iron less available to micro-organisms that inhibit their growth [86]. Regulation of systemic iron metabolism, including organs and cell types involved in systemic iron balance are discussed in details [86].

For example, cells (such as enterocytes) are absorbing “dietary iron” by divalent metal transporter 1 “located on the apical surface upon reduction of Fe^3+^ to Fe^2+^” by ferrireductases such as duodenal cytochrome B [86]. These cells in addition to the macrophages (“spleenic reticuloendothelial”) that are recycling iron from “senescent red blood cells” [86], finally “release iron via ferroportin with the aid of hephaestin, which oxidizes of Fe^3+^ to Fe^2+^” [86]. Circulating plasma transferrin (which represents “the most dynamic body iron pool”) is transferring this metal all around the animal body. Another conditions “that affect iron metabolism indirectly” are the following: “inflammation, ER stress, erythropoiesis, and hypoxia” [86]. 

### 3.9. Lead (Plumbum)

A significant increase in the content of Pb in animals with lesions of the cardiovascular system and a tendency to an increase in the content of metals in lesions of the excretory system and reproductive organs were revealed. Lead has a negative impact on health [47], which is associated with a pronounced toxicity of the metal for a number of systems and organs [87] and primarily for the nervous system [88]. The influence of lead on the maternal organism and the development of congenital heart anomalies in newborns [89], as well as congenital neural tube defects [90], has been established, which is associated with the ability of MME to disrupt the regulatory mechanisms of DNA methylation [91,92]. Scientists note the role of Pb in the development of oncological diseases, but these data were not confirmed in our study [93]. It is well-known [84,94] that Pb is able to influence metabolic processes.

### 3.10. Cadmium

The sample found a significant increase in the content of cadmium in animals with diseases of the cardiovascular and circulatory systems. The hematopoietic system is one of the targets of the toxic action of Cd. It has been shown that an increase in the level of the xenobiotic in the blood is caused by a decrease in the concentration of hemoglobin [95] and increases the likelihood of developing iron deficiency anemia [96,97]. Experiments have shown that exposure to cadmium is accompanied by a shift in the blood formula towards myelopoiesis [98], as well as hemolysis and insufficient production of erythropoietin [99,100].

### 3.11. Arsenic

A significant increase in the content of As was found in the presence of diseases of the excretory system. Arsenic induces the formation of oxidized lipids, which in turn generate several bioactive molecules (ROS, peroxides, and isoprostanes), the main end products of which are aldehydes. There is an indication of chronic and acute exposure to As in the etiology of cancer, cardiovascular disease (hypertension and atherosclerosis), neurological disorders, gastrointestinal disorders, liver and kidney disease, reproductive health effects, skin changes, and other health disorders. The role of antioxidant defense systems against arsenic toxicity is also discussed in detail [101].

## 4. Conclusions

The level of MME accumulation in the body of animals can be the key reason to the occurrence of microelementoses, i.e., the development of various diseases. A study of the “nosological profile” of diseases was carried out and it was found that up to 12.9% of the animals from the total number of livestock (from four Russian zoological institutions) were exposed to diseases. The following tendency of disease frequency for all animals in our research was obtained: digestive system > musculoskeletal system ≥ cardiovascular system > hearing organs ≥ nervous system. During the research, an increase in the proportion of oncological diseases in all the studied institutions was established. A significant relationship was established between the accumulation of Zn and the following diseases: skin, digestive, and vision systems, as well as oncological diseases, Cu—with diseases of the musculoskeletal and the cardiovascular system, oncological diseases, Fe—with diseases of the cardiovascular system, Pb—with metabolic diseases, nervous and excretory systems, oncological diseases, Cd—with diseases of the cardiovascular and nervous system, and As—with diseases of the excretory systems. There was a tendency of a decrease in the concentration of MMEs in case of metabolic disorders and diseases of the reproductive system. Therefore, monitoring of the MME status of the organism of all animals must be carried out regularly (optimally—once every 6 months).

## Figures and Tables

**Figure 1 animals-13-00852-f001:**
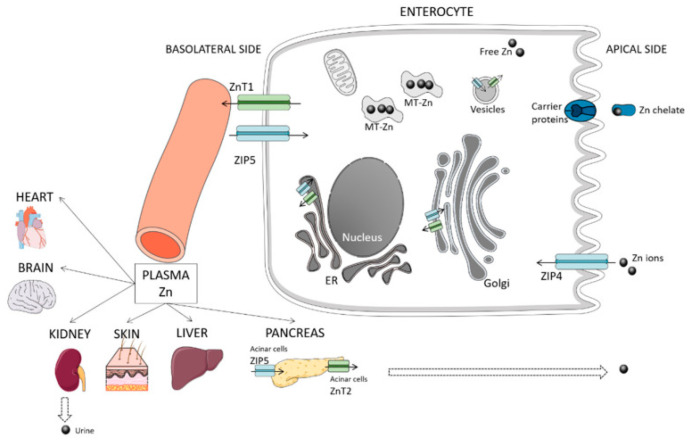
Simplified scheme of zinc metabolism: absorption and excretion from the intestine: ZIPS, ZIP4, ZnT1, and ZnT2—transporters of enterocyte membranes and intracellular components (shown in blue and green, respectively); excretion routes are shown by dotted arrows; ER—endoplasmic reticulum; MT—metallothionein; some organs (heart, brain, kidney, skin, liver, pancreas) that exchange zinc with the plasma pool are presented (adapted from [63]).

**Figure 2 animals-13-00852-f002:**
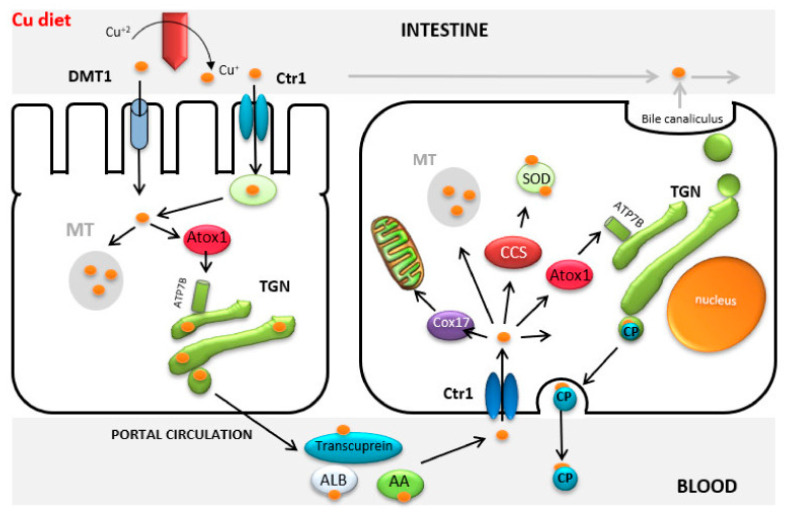
Copper transport. Ctr1: Copper specific transporter; DMT1: Divalent Metal Transporter 1; MT: Metallothionein; TGN: Trans Golgi Network; Cox17 and Atox1: Copper chaperone proteins; SOD: Superoxide dismutase; CP: Ceruloplasmin; CCS: Cytochrome c oxidase; ATP7B: Copper-transporting P-type ATPase (adapted from [76]).

**Table 1 animals-13-00852-t001:** Correlation between the MME content (mg/kg) in samples of derived integument with general morbidity of certain classes of diseases (with a proven role of environmental factors).

Group of Nosologies	MME, mg/kg					
	Zn	Cu	Fe	Pb	Cd	As
Skin diseases	**0.43 ***	0.19	0.41	0.72 *	0.08	−0.04
Diseases of the hearing organs	−0.14	0.24	0.34	−0.11	0.17	0.01
Diseases of the musculoskeletal system	0.43	**0.24 ***	0.51	0.41	0.12	0.05
Diseases of the respiratory system	−0.26	0.14	0.11	0.01	0.34	0.29
Diseases of the digestive system	0.38	0.12	0.10	0.62	0.23	0.11
Metabolic diseases	0.03	−0.05	0.12	**0.27 ***	0.18	0.11
Diseases of the cardiovascular system	−0.01	**0.37 ***	0.24	0.63	**0.18 ***	0.16
Diseases of the organs of vision	0.42	0.22	0.41	0.22	−0.21	0.08
Diseases of the nervous system	0.02	0.11	0.09	**0.11 ***	−0.24	0.23
Diseases of the reproductive system	0.21	0.17	0.13	−0.08	−0.01	0.41
Diseases of the excretory system	0.11	0.04	0.04	**0.34 ***	−0.01	0.01
Oncological diseases	**0.28 ***	0.19	**0.11 ***	**0.03**	0.17	0.15

* *p* < 0.05.

**Table 2 animals-13-00852-t002:** The MME level (mg/kg) in animal biological media, taking into account past disease.

Group of Nosologies	ME, mg/kg					
	Zn	Cu	Fe	Pb	Cd	As
Skin diseases	**56.9 ± 8.4 *** ↓	15.9 ± 2.45	543 ± 113	6.99 ± 2.48	1.01 ± 0.48	0.110 ± 0.102
Diseases of the hearing organs	129 ± 11.2	11.7 ± 14.5	428 ± 48.6	8.01 ± 4.24	1.52 ± 0.93	0.211 ± 0.071
Diseases of the musculoskeletal system	143 ± 2.42	**3.95 ± 4.57 *** ↓	395 ± 38.5	7.11 ± 3.24	3.15 ± 1.01	0.342 ± 0.241
Diseases of the respiratory system	166 ± 3.84	13.8 ± 9.42	288 ± 69.0	6.99 ± 2.97	1.13 ± 0.24	0.533 ± 0.042
Diseases of the digestive system	**62.1 ± 6.51 *** ↓	12.5 ± 7.15	343 ± 57.5	6.97 ± 2.11	1.56 ± 0.64	0.241 ± 0.112
Metabolic diseases	74.5 ± 19.5 ↓	14.6 ± 4.52	354 ± 15.8	8.01 ± 3.54	1.85 ± 1.03	0.653 ± 0.431
Diseases of the cardiovascular system	142 ± 54.9	18.7 ± 3.54 ↑	**128 ± 11.8 *** ↓	6.88 ± 1.02	**5.01 ± 0.68 *** ↑	0.342 ± 0.123
Diseases of the organs of vision	**49.7 ± 33.8 *** ↓	14.7 ± 8.41	302 ± 65.8	7.55 ± 1.24	2.03 ± 1.51	0.429 ± 0.221
Diseases of the nervous system	169 ± 64.5	11.6 ± 5.64	342 ± 54.2	**14.6 ± 2.94 *** ↑	**3.48 ± 0.86 *** ↑	0.538 ± 0.341
Diseases of the reproductive system	98.8 ± 29.6 ↓	11.9 ± 8.31	298 ± 51.4	11.68 ± 1.06 ↑	1.56 ± 1.02	0.668 ± 0.339
Diseases of the excretory system	146 ± 25.4	11.7 ± 5.42	270 ± 185	12.97 ± 2.84 ↑	2.11 ± 0.63	**0.951 ± 0.022 *** ↑
Oncological diseases	155 ± 49.8	**21.9 ± 2.48 *** ↑	298 ± 67.8	9.01 ± 2.48	2.46 ± 1.39	0.638 ± 0.109
Average level	130 ± 14.3	17.7 ± 4.89	388 ± 48.8	5.73 ± 0.93	1.25 ± 0.17	0.768 ± 0.181

* *p* < 0.05.

## Data Availability

The research data obtained during this study are subject of special protection (particular privacy restrictions of ROSBIOTECH), i.e. not available to public.

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
