# Peer review of "Relationships between the Content of Micro- and Macroelements in Animal Samples and Diseases of Different Etiologies"

_animals, 2023, doi:10.3390/ani13050852_

Round 1
Reviewer 1 Report
This Manuscript suffers some major fails in the disposition, terminology, writing style, readability and fluency. Nonetheless, the manuscript could be much improved and could be a useful contribution to our knowledge on the subject and can be accepted after these minor revisions:
1- English should improve by a native person. The paper suffers from a poor English structure throughout and cannot be published or reviewed properly in the current format. The manuscript requires a thorough proofread by a native person whose first language is English. The instances of the problem are numerous and this reviewer cannot individually mention them. It is the responsibility of the author(s) to present their work in an acceptable format. Unless the paper is in a reasonable format, it should not have been submitted.
2- The novelty of the study needs to be highlighted compare to other similar studies or consider to explicitly mention what is gap knowledge and/or what was lacking in the indicated studies.
3- Discussion is weak. The discussion needs enhancement with real explanations not only agreements and disagreements. Authors should improve it by the demonstration of biochemical/physiological causes of obtained results. Instead of just justifying results, results should be interpreted, explained to appropriately elaborate inferences. Discussion seems to be poor, didn't give good explanations of the results obtained. I think that it must be really improved. Where possible please discuss potential mechanisms behind your observations. You should also expand the links with prior publications in the area, but try to be careful to not over-reach. For the latter, you should highlight potential areas of future study.
4- The scientific background of the topic is poor. In "Introduction" and "Discussion", the authors should cite recent references between 2019-2022 from JCR journals (with impact factor) about recent achievements on the element feeding in poultry. For example, authors should cite to:
Navidshad B., Mohammadrezaei M., Zarei M., Valizadeh R., Karamati S., Rezaei F., Jabbari S., Kachoei R. and Esmaeilinasab P. (2019). The new progresses in trace mineral requirements of broilers, a review. Iranian J. Appl. Anim. Sci. 9(1), 9-16.
Faghih-Mohammadi, F., Seidavi, A. and Bouyeh, M. 2022. The effects of chelated micro-elements feeding in broiler breeder hens and their progeny: A review. Tropical Animal Health and Production. 54(323): 1-14. doi: 10.1007/s11250-022-03317-1
5- A detailed "Conclusion" should be provided to state the final result that the authors have reached. Please note you only need to place your conclusion and not keep putting results, because these have already been presented in the manuscript.
6- Author(s) should re-format the references based on journal format. See the instructions for authors.
7- The numbers and decimals in Tables should be follow the rule of: xxxx, xxx, xx.x, x.xx, 0.xxx and 0.0xxx
Author Response
Answers to Reviewer 1
Reviewer Comment 1:
1- English should improve by a native person. The paper cannot be published or reviewed properly in the current format. The manuscript requires a thorough proofread by a native person whose first language is English. It is the responsibility of the author(s) to present their work in an acceptable format. Unless the paper is in a reasonable format, it should not have been submitted.
Answers to Reviewer Comment 1
- Authors have tried to use a proofread by the English Editing Service by “Animals”, but failed with payment from Russia to Switzerland. That is why we have used a help of our colleagues in the Department of the Foreign Languages (English).
Reviewer Comment 2:
2- The novelty of the study needs to be highlighted compare to other similar studies or consider to explicitly mention what is gap knowledge and/or what was lacking in the indicated studies.
Answers to Reviewer Comment 2
- Authors added some recent papers and highlighted the “gap knowledge” in the text.
Reviewer Comment 3:
3- Discussion is weak. The discussion needs enhancement with real explanations not only agreements and disagreements. Authors should improve it by the demonstration of biochemical/physiological causes of obtained results. Instead of just justifying results, results should be interpreted, explained to appropriately elaborate inferences. Discussion seems to be poor, didn't give good explanations of the results obtained. I think that it must be really improved. Where possible please discuss potential mechanisms behind your observations. You should also expand the links with prior publications in the area, but try to be careful to not over-reach. For the latter, you should highlight potential areas of future study.
Answers to Reviewer Comment 1
- Authors rewrote some parts in the "Discussion" section with explanations of the obtained results. Authors added some recent papers and Figures 1-3 with simplified schemes of Zn, Cu and Fe metabolisms.
Reviewer Comment 4:
4- The scientific background of the topic is poor. In "Introduction" and "Discussion", the authors should cite recent references between 2019-2022 from JCR journals (with impact factor) about recent achievements on the element feeding in poultry. For example, authors should cite to:
Navidshad B., Mohammadrezaei M., Zarei M., Valizadeh R., Karamati S., Rezaei F., Jabbari S., Kachoei R. and Esmaeilinasab P. (2019). The new progresses in trace mineral requirements of broilers, a review. Iranian J. Appl. Anim. Sci. 9(1), 9-16.
Faghih-Mohammadi, F., Seidavi, A. and Bouyeh, M. 2022. The effects of chelated micro-elements feeding in broiler breeder hens and their progeny: A review. Tropical Animal Health and Production. 54(323): 1-14. doi: 10.1007/s11250-022-03317-1
Answer to Reviewer Comment 4:
- Authors rewrote the scientific background of the topic. Authors rewrote the parts "Introduction" and "Discussion" and cited recent references between 2019-2022 from JCR journals (with impact factor) and book (about recent achievements on the micro- and macroelements in animal tissues), including the element feeding in poultry (ref.22. Navidshad B., Mohammadrezaei M., Zarei M., Valizadeh R., Karamati S., Rezaei F., Jabbari S., Kachoei R. and Esmaeilinasab P. (2019). The new progresses in trace mineral requirements of broilers, a review. Iranian J. Appl. Anim. Sci. 9(1), 9-16. and ref.23. Faghih-Mohammadi, F., Seidavi, A. and Bouyeh, M. 2022. The effects of chelated micro-elements feeding in broiler breeder hens and their progeny: A review. Tropical Animal Health and Production. 54(323): 1-14. doi: 10.1007/s11250-022-03317-1).
Reviewer Comment 5:
- A detailed "Conclusion" should be provided to state the final result that the authors have reached. Please note you only need to place your conclusion and not keep putting results, because these have already been presented in the manuscript.
Answer to Reviewer Comment 5:
- Authors rewrote the "Conclusion" in the recommended way.
Reviewer Comment 6:
- Author(s) should re-format the references based on journal format. See the instructions for authors.
Answer to Reviewer Comment 6:
- Authors “re-formated the references based on journal format” (using the instructions for authors).
Reviewer Comment 7:
- The numbers and decimals in Tables should be follow the rule of: xxxx, xxx, xx.x, x.xx, 0.xxx and 0.0xxx
Answer to Reviewer Comment 7:
- Authors corrected the numbers and decimals in Table 2 in the style of: xxxx, xxx, xx.x, x.xx, 0.xxx and 0.0xxx. We decided to leave the style of the numbers in Table 1, because the data of the correlations usually presented as the following:
±0.75 - ±1.0 = very strong positive (negative) correlations;
±0.50 - ±0.74 = strong positive (negative) correlations ;
±0.25 - ±0.49 = middle positive (negative) correlations;
±0.01 - ±0.24 = weak positive (negative) correlations.

Reviewer 2 Report
Marina V. Stepanova, Larisa F. Sotnikova, Sergei Yu. Zaitsev Relationships of the Content of Micro- and Macroelements in Animal Samples and the Diseases of Different Etiologies The article focuses on zoo animals and the role of macro and trace elements as their health. It is of great practical importance for employees of various zoos and nurseries of rare species of animals. I recommend it for publication. It is written in a competent scientific language, easy to read. I have only a positive opinion about the possibility of its publication in your magazine.
Author Response
Dear Reviewer,
Authors are thankful to you for the positive review and comments. We have tried to use a proofread by the English Editing Service by “Animals”, but failed with payment from Russia to Switzerland. That is why we have used a help of our colleagues in the Department of the Foreign Languages (English).
Authors rewrote some parts in the "Discussion" section with explanations of the obtained results. Authors added some recent papers and Figures 1-3 with simplified schemes of Zn, Cu and Fe metabolisms.
Authors rewrote the scientific background of the topic as mentioned by Reviewer 1. Authors rewrote the parts "Introduction" and "Discussion" and cited recent references between 2019-2022 from JCR journals (with impact factor) and book (about recent achievements on the micro- and macroelements in animal tissues), including the element feeding in poultry (ref.22. Navidshad B., Mohammadrezaei M., Zarei M., Valizadeh R., Karamati S., Rezaei F., Jabbari S., Kachoei R. and Esmaeilinasab P. (2019). The new progresses in trace mineral requirements of broilers, a review. Iranian J. Appl. Anim. Sci. 9(1), 9-16. and ref.23. Faghih-Mohammadi, F., Seidavi, A. and Bouyeh, M. 2022. The effects of chelated micro-elements feeding in broiler breeder hens and their progeny: A review. Tropical Animal Health and Production. 54(323): 1-14. doi: 10.1007/s11250-022-03317-1).
Authors corrected the numbers and decimals in Table 2 in the style of: xxxx, xxx, xx.x, x.xx, 0.xxx and 0.0xxx (suggested by Reviewer 1). We decided to leave the style of the numbers in Table 1, because the data of the correlations usually presented as the following:
±0.75 - ±1.0 = very strong positive (negative) correlations;
±0.50 - ±0.74 = strong positive (negative) correlations ;
±0.25 - ±0.49 = middle positive (negative) correlations;
±0.01 - ±0.24 = weak positive (negative) correlations.

Round 2
Reviewer 1 Report
-